# Multiparametric Evaluation of *Tetradesmus obliquus* Biomass: An Integrated Approach Including Antioxidant, Nutritional, and Energy Properties

**DOI:** 10.3390/microorganisms13071583

**Published:** 2025-07-04

**Authors:** Gilvana Scoculi de Lira, Ihana Aguiar Severo, Fernando Augusto Ferraz, Iago Gomes Costa, Matheus Murmel Guimarães, Ingrid Fátima Zattoni, Luiz Fernando Bianchini, José Viriato Coelho Vargas, Dhyogo Miléo Taher, André Bellin Mariano

**Affiliations:** 1Graduate Program in Engineering and Material Science, Federal University of Paraná (PIPE/UFPR), Curitiba 81531-980, Brazil; iago_gomes_costa@hotmail.com (I.G.C.); viriato@ufpr.br (J.V.C.V.); dhyogo@ufpr.br (D.M.T.); andrebmariano@ufpr.br (A.B.M.); 2Sustainable Energy Research and Development Center, Federal University of Paraná (NPDEAS/UFPR), Curitiba 81531-990, Brazil; 3Department of Mechanical Engineering, FAMU-FSU College of Engineering, Energy and Sustainability Center, Center for Advanced Power Systems (CAPS), Florida A&M University-Florida State University, Tallahassee, FL 32310-6046, USA; 4Forest Engineering and Technology Department, Federal University of Paraná (DETF/UFPR), Curitiba 80210-170, Brazil; fernando.ferraz@ufpr.br; 5School of Medicine and Life Sciences, Pontifical Catholic University of Paraná (PUC-PR), Curitiba 80215-901, Brazil; murmel.guimaraes@pucpr.br (M.M.G.); ingrid.zattoni@pucpr.br (I.F.Z.); fernando.bianchini@pucpr.br (L.F.B.); 6Department of Mechanical Engineering, Federal University of Paraná (DEMEC/UFPR), Curitiba 81531-970, Brazil; 7Department of Electrical Engineering, Federal University of Paraná (DELT/UFPR), Curitiba 81530-000, Brazil

**Keywords:** calorific value, industrial photobioreactors, microalgal biomass, bioproducts, physicochemical composition, antioxidant, protein content

## Abstract

The microalga *Tetradesmus obliquus* has emerged as a promising candidate for biotechnological and industrial applications due to its rapid growth, resilience under diverse environmental conditions, and potential for bioactive compound production. This study presents a multiparametric characterization of dry *T. obliquus* biomass cultivated in patented industrial-scale photobioreactors, integrating thermochemical, elemental, antioxidant, and protein analyses. Proximate and ultimate analyses were conducted to assess fuel potential, revealing favorable volatile matter (VM = 64.80–72.44%) and fixed carbon (FC = 15.77–21.23%) contents. The HHV (18.32–22.75 MJ·kg^−1^) and LHV (16.86–21.24 MJ·kg^−1^) confirmed the biomass as a viable candidate for solid biofuel. The elemental composition provided the total nitrogen values, subsequently used to estimate the protein content via both the Kjeldahl and Dumas methods, with results ranging from 36.66% to 40.02%, in line with the literature. Despite the absence of detectable antioxidant activity under the tested DPPH conditions, the biomass demonstrated a robust nutritional profile and energy potential. These findings support the industrial relevance of *T. obliquus* biomass, particularly for applications targeting sustainable protein sources and bioenergy solutions.

## 1. Introduction

Microalgae are unicellular, eukaryotic, and photosynthetic organisms that have gained increasing attention due to their ability to produce biomass rich in bioactive compounds such as proteins, lipids, carbohydrates, and antioxidants. These characteristics make them promising candidates for applications in the food, pharmaceutical, nutraceutical, cosmetic, and biofuel industries, owing to their high contents of proteins, essential amino acids, fatty acids, pigments, and other functional metabolites. Additionally, their rapid growth rates and capacity for cultivation under controlled conditions contribute to both the sustainability and efficiency of biomass production [1].

The genus *Tetradesmus obliquus* (Turpin) M.J. Wynne, previously classified under the genus *Scenedesmus*, is a green microalga widely distributed in freshwater environments such as lakes and rivers. This species is particularly valued for its high lipid accumulation, ease of cultivation, and tolerance to a wide range of temperatures and pH levels. Moreover, *T. obliquus* is effective in removing heavy metals, nitrogen, and phosphorus from aquatic systems and displays resilience under high carbon dioxide (CO_2_) concentrations, resulting in high biomass productivity. These attributes make it a strong candidate for applications in bioenergy and wastewater treatment [2,3].

The cultivation of microalgae in industrial photobioreactors enables improvement in growth parameters, including light intensity, temperature, and nutrient availability, leading to enhanced productivity and biomass quality. This approach also facilitates process standardization and scalability for commercial applications [4].

Several microalgal species have demonstrated significant antioxidant activity, primarily attributed to the presence of compounds such as pigments (e.g., carotenoids), polyphenols, and vitamins. These natural antioxidants are of particular interest to the food and pharmaceutical industries, providing viable alternatives to synthetic antioxidants. The evaluation of antioxidant capacity using assays such as DPPH (2,2-diphenyl-1-picrylhydrazyl) and ABTS (2,2′-azino-bis(3-ethylbenzothiazoline-6-sulphonic acid)) is essential for detecting and quantifying such bioactive compounds [5].

Beyond its wide industrial relevance and role in sustainable production systems, *T. obliquus* is particularly noted for its diverse biological activities, especially its ability to synthesize bioactive compounds with therapeutic potential. This species is a rich source of essential fatty acids (EFAs), including polyunsaturated fatty acids (PUFAs), which are crucial for maintaining human health and managing various medical conditions [6].

As these fatty acids cannot be synthesized endogenously, *T. obliquus* represents a valuable dietary alternative. Moreover, this microalga is abundant in amino acids essential for protein biosynthesis. Under mixotrophic cultivation, it demonstrates significantly enhanced protein yields compared to under conventional cultivation methods. Its ability to regulate metabolic pathways and improve amino acid profiles further supports its potential use in the food and animal feed industries, where its biomass may serve as a high-value nutritional input [7].

Antioxidants are critical molecules that protect cells from oxidative damage caused by free radicals—highly reactive species generated during normal metabolism or in response to environmental factors such as pollution, radiation, and smoking. Free radical accumulation induces oxidative stress, damaging lipids, proteins, and DNA, and contributing to cellular ageing and the development of chronic diseases, including cancer, cardiovascular disorders, and neurodegenerative conditions. Recent studies have shown that *T. obliquus*, a green microalga, exhibits strong antioxidant potential due to its content of phenolic compounds, flavonoids, and pigments such as carotenoids. Additionally, it demonstrates noteworthy antitumor and antimicrobial activities [8].

In parallel, the nutritional profile of this microalga is enhanced by its high protein concentration, with soluble protein levels exceeding 28% of its dry biomass [7], in addition to its capacity to stabilize food emulsions as a functional protein source. The combination of antioxidant activity and nutritional value positions *T. obliquus* as a strong candidate for incorporation into functional formulations within the food, pharmaceutical, and cosmetic industries [9]. These synergistic characteristics further support its potential use across a wide range of industrial sectors.

Among microalgal species with notable nutritional value, *T. obliquus* stands out for its high protein content, with values ranging from 50% to 56% of its dry biomass, positioning it among the richest known microalgal protein sources. This profile is comparable to, or even exceeds, traditional protein sources such as skimmed milk powder (36%) and beef (43%). Its nutritional profile has been highlighted as particularly relevant for sustainable diets, especially in contexts requiring alternatives to animal-derived proteins, thereby expanding its potential in functional food formulations and nutritional supplements. The versatility of *T. obliquus* biomass is attributed not only to its protein composition but also to its content of micronutrients, fatty acids, and antioxidant compounds, reinforcing its role as a promising functional ingredient for a wide range of industrial applications [10].

Realizing the biotechnological potential of microalgae for commercial applications requires overcoming key challenges related to large-scale cultivation. Although microalgae possess vast potential, their commercial viability still hinges on the efficient transition from laboratory-scale processes to pilot- and industrial-scale operations. This transition is particularly critical in the context of photobioreactors (PBRs), which provide tightly controlled conditions for microalgal growth. PBRs allow for the precise regulation of variables such as light intensity, temperature, and nutrient concentrations, thereby improving biomass production. Additionally, these systems offer key logistical advantages, including reduced land area requirements and higher productivity, with shorter and more efficient cultivation cycles [11,12].

Given this context, the present study aimed to perform a comprehensive multiparametric characterization of *T. obliquus* biomass, focusing on its physicochemical, nutritional, thermochemical, and antioxidant properties. This characterization was conducted to evaluate its potential as an innovative ingredient for industrial applications, particularly within the food, pharmaceutical, and bioenergy sectors. To this end, dry biomass samples produced by the Sustainable Energy Research and Development Center (NPDEAS) research group were analyzed.

The biomass was cultivated in a patented industrial-scale PBR system integrated with a residue incinerator, enabling large-scale production under controlled engineering conditions. This sustainable production process aligns with several United Nations Sustainable Development Goals (SDGs), including Goal 2—Zero Hunger; Goal 3—Good Health and Well-being; Goal 7—Affordable and Clean Energy; Goal 9—Industry, Innovation and Infrastructure; Goal 11—Sustainable Cities and Communities; Goal 12—Responsible Consumption and Production; and Goal 13—Climate Action.

## 2. Materials and Methods

### 2.1. Samples

The dried microalgal biomass was obtained from processes carried out by the NPDEAS research group, specifically following the methodology proposed by Costa et al. (2024) [12] and Costa et al. (2022) [13]. Microalgae were cultivated using two types of culture media: a synthetic CHU medium [14] and digested swine effluent diluted to 10%. The latter was identified as the most effective dilution [15].

Microalgae production started at a laboratory scale and was then gradually scaled up to a pilot scale using 12 L airlift PBRs and later to an industrial scale with 12 m^3^ PBRs [11,12,13], as presented in Figure 1.

The species *Tetradesmus obliquus* was grown for 15 days in industrial PBRs with a working volume of 12 m^3^. Following the cultivation period, the cultures were transferred to flocculation tanks for biomass recovery. To facilitate separation, the flocculant Tanfloc SG was applied, and the culture pH was adjusted to 7 using industrial-grade CO_2_, ensuring optimal flocculant performance. The flocculation process involved 15 min of agitation, followed by an overnight settling phase to allow for effective biomass sedimentation. The recovered biomass was then concentrated via centrifugation using a US Centrifuge System M512, operated at 3000 rpm and a flow rate of 4 L·min^−1^ [12].

Following centrifugation, the resulting microalgal paste was uniformly spread onto trays to undergo the drying process. A variety of drying techniques were employed, including (i) open-air drying, (ii) vacuum drying, (iii) spray drying, (iv) drum drying, (v) freeze drying, (vi) fluidized bed drying, and (vii) drying in temperature-controlled ovens. Lipid extraction was carried out using organic solvents, and lipid compounds were subsequently separated through distillation [12,13].

The dried crude extract (lipid fraction) and the residual biomass (after lipid removal) were then ground using a knife mill to obtain a fine microalgal powder with a 60-mesh particle size. Figure 2 provides a concise schematic representation of the procedures employed for sample processing and biomass extraction, illustrating the key steps involved from drying to lipid separation and biomass grinding.

Following the completion of biomass processing and extraction procedures, the resulting samples were categorized and characterized. These samples served as the basis for all subsequent analyses carried out in this study. A detailed description of each sample is presented in Table 1.

### 2.2. Physicochemical Characterization

For the physicochemical characterization of the dried microalgal biomass, the following analyses were performed: proximate analysis, calorific value determination, and ultimate analysis.

Proximate Analysis

Proximate analysis is a fundamental method for characterizing biomass, encompassing the quantification of the moisture content (M), volatile matter (VM), ash content (A), and fixed carbon (FC). It was performed in accordance with the American society for Testing and Materials, ASTM D-1762/84 [16].

To determine the ash content, approximately 1 g of low-moisture sample was weighed into a pre-weighed crucible and placed in a muffle furnace at 600 ± 10 °C until complete combustion of the material. The crucible was then removed from the furnace and cooled in a desiccator until it reached room temperature. Finally, the ash content was calculated using Equation (1):(1)$$A=\frac{m_{1}-m_{0}}{m}\times 100$$
whereA = ash content (%);$m_{0}$ = mass of the empty crucible (g);$m_{1}$ = mass of the crucible plus residue sample (g);m = mass of the sample (g).

To determine VM, approximately 1 g of pre-dried sample was weighed into a pre-tared crucible and placed in a muffle furnace at 900 ± 10 °C for 7 min. After this period, the crucible was removed from the furnace and cooled in a desiccator prior to final mass determination. The volatile matter content was then calculated using Equation (2):(2)$$VM=\frac{m_{2}-m_{3}}{m}\times 100$$
whereVM = volatile matter (%);$m_{2}$ = initial mass of crucible + residue sample (g);$m_{3}$ = final mass of crucible + residue sample (g);m = mass of the residue sample (g).

The FC content corresponds to the remaining mass after the release of volatile compounds and is obtained by subtracting the values of ash and moisture. Equation (3) describes the calculation of fixed carbon:(3)$$\mathrm{FC}=100-\left(A\;+\;\mathrm{VM}\right)$$
whereFC = fixed carbon (%);A = ash content (%);VM = volatile matter (%).
Calorific Value

The calorific value, both the higher heating value (HHV) and lower heating value (LHV), indicates the amount of energy potentially released by the biomass during combustion. The HHV includes the heat released from the condensation of water formed during combustion, whereas the LHV excludes this latent energy. The HHV was determined using an IKA C-5000 bomb calorimeter, following the guidelines of ISO 18125:2017 [17]. The LHV was calculated based on hydrogen content data obtained from the literature, in accordance with the same standard.
Ultimate Analysis

The ultimate analysis of the biomass determines the chemical composition of the fuel, specifically the contents of carbon (C), hydrogen (H), nitrogen (N), and oxygen (O), the latter being calculated by difference. This composition provides essential information for the combustion process [18,19]. The analyses were performed using a Perkin Elmer 2400 Series II CHN-O elemental analyzer.

### 2.3. Antioxidant Activity Evaluation Using the DPPH Assay

To evaluate the antioxidant activity of the microalgal biomass, the 2,2-diphenyl-1-picrylhydrazyl (DPPH) free radical method was employed, adapted from the protocol described by Huang et al. (2022) [20], which is widely used in antioxidant assays for plant-based samples. The main steps of the adapted methodology are described below.

Calibration Curve: Mixtures containing 3 mL of five methanolic solutions of DPPH (Sigma-Aldrich, St. Louis, MO, USA) and the positive control Trolox (±)-6-hydroxy-2,5,7,8-tetramethylchromane-2-carboxylic acid—a water-soluble analogue of vitamin E (Sigma-Aldrich, St. Louis, MO, USA)—were prepared. The final DPPH concentration was fixed at 0.02 mM, while Trolox was added at varying concentrations (0.0075, 0.01, 0.015, 0.02, and 0.03 mM). The mixtures were homogenized by agitation and incubated in the dark for 5 min. Absorbance readings were then performed using a spectrophotometer at a wavelength of 517 nm. All measurements were carried out in triplicate. Based on the absorbance values and Trolox concentrations, a standard calibration curve was generated.

Antioxidant Activity Assays: The assays were conducted by adding 100 mg of each of the six microalgal biomass samples (solid fraction) to 100 mL of methanol. The mixtures were filtered to obtain the methanol-soluble fractions, from which 0.2 mL aliquots were combined with 2.8 mL of a methanolic DPPH solution (final DPPH concentration: 0.02 mM). The solutions were homogenized by agitation and incubated in the dark for 5 min. Absorbance was then measured at 517 nm using a spectrophotometer. Reductions in absorbance, indicative of antioxidant activity, were expressed in Trolox equivalents (TE), based on the calibration curve. A schematic overview of the experimental steps is shown in Figure 3.

### 2.4. Protein Content Determination

Kjeldahl Method: The Kjeldahl method is one of the most traditional procedures for determining total nitrogen in organic matrices. The nitrogen determination approach proposed by Kjeldahl is widely applied for protein analysis in various food and biological matrices. Accordingly, this method was used to assess the protein content of the microalgal biomass [21].

The method is based on the digestion of the sample with concentrated sulfuric acid, which promotes the conversion of organic nitrogen into ammonia, followed by distillation and titration. Although the Kjeldahl method is widely applied, it does not distinguish between protein and non-protein nitrogen. Therefore, the selection of an appropriate nitrogen-to-protein conversion factor is crucial for accurately estimating the true protein content of the biomass under study. The conversion factor (k) adopted in this study was based on the work of Templeton and Laurens (2015) [22], which proposed a factor of 5.08 for species belonging to the genera *Scenedesmus*/*Tetradesmus*. The main steps of the adapted methodology are described below:Moisture content determination: Mettler Toledo HE53. Since the presence of moisture in the sample can affect its weight and lead to inaccurate results, it is essential to determine the moisture content in order to calculate the nitrogen content based on the dry mass of the sample [23,24]. Accordingly, the moisture content was first determined using a Mettler Toledo HE53 moisture analyzer.

Reagents: 10 mL of concentrated sulfuric acid; 50 mL of 0.1 M hydrochloric acid (with correction factor applied); 40 mL of 50% sodium hydroxide; 40 mL of 4% boric acid; 2.5 g of catalytic mixture (comprising 18 g of sodium selenite, 20 g of copper sulphate, and 242.5 g of potassium sulphate); and 0.5 mL of mixed indicator solution (0.132 g of methyl red and 0.066 g of bromocresol green dissolved in 200 mL of ethanol).
Digestion: This step consists of weighing 0.2 g of the solid sample and transferring it to a digestion tube. Then, 10 mL of concentrated sulfuric acid and 2.5 g of the catalytic mixture are added, as shown in Figure 4.

The digestion was carried out using a TECNAL TE-152—SCRUBBER digester. The tube was placed on the digestion plate, and heating was initiated at 100 °C. The temperature was then increased gradually by 50 °C every 10 min until it reached 350 °C, at which point a greenish color developed in the solution. The system was then turned off and allowed to cool. Figure 5 illustrates the digestion process.
Distillation: To begin the process, distilled water was added to the digestion tube until the volume was approximately doubled. The tube was then connected to the distillation unit, ensuring that all fittings were properly aligned. Next, 50% sodium hydroxide (NaOH) was added until the solution turned dark in color, indicating appropriate alkalinization for ammonia release. The heating level was set to position 6, and the cooling system was activated simultaneously. The ammonia released during the process was captured in an Erlenmeyer flask containing 40 mL of 4% boric acid solution and 0.5 mL of mixed indicator. Distillation was maintained for 10 min or until the solution in the Erlenmeyer flask displayed a stable green color, indicating the complete capture of the ammonia. This distillation step was carried out using a TECNAL TE-0364 nitrogen distiller, as shown in Figure 6.


Titration: The titration step began with the filling of the burette using 0.1 M hydrochloric acid, which had been previously standardized. The distillate obtained from the previous step was titrated until a color change from green to pink was observed, indicating the endpoint of the reaction, as shown in Figure 7. The volume of hydrochloric acid consumed during titration was recorded for the subsequent calculation of the total nitrogen content in the analyzed samples.


To calculate the protein content, the total nitrogen content (%N) was first determined based on the volume of standardized hydrochloric acid consumed during titration, using Equation (4):(4)$$\%N=\frac{\left(V_{sample\;tritation\;}-V_{blank}\right)x\;\left[HCl\right]\times 14}{dry\;sample\;mass}\times 100$$
where$V_{sample\;tritation}$ in L;$V_{blank}$ in L;[*HCl*] in mol L^−1^;14 = molar mass of nitrogen (g mol^−1^);dry sample mass in g.

The protein content was estimated by applying a specific nitrogen-to-protein conversion factor of 5.08, as recommended by Templeton and Laurens (2015) [22] for microalgae of the genus *Tetradesmus*. This calculation was performed using the percentage values of organic nitrogen obtained from the Kjeldahl method, using Equation (5):(5)%Protein=%N×5.08

Dumas Method: This is an established technique for quantifying the total nitrogen content in organic samples through high-temperature combustion in a CHN elemental analyzer, which is subsequently converted to crude protein content using an appropriate nitrogen-to-protein conversion factor. In this method, a homogenized sample is combusted at high temperatures in an oxygen-enriched environment, resulting in the oxidation of nitrogenous compounds to nitrogen oxides. These oxides are then reduced to elemental nitrogen (N_2_), which is quantified using a thermal conductivity detector [25]. Thus, for protein determination, the total nitrogen value obtained from elemental analysis was used, applying Equation (5).

All experimental data obtained from the different analyses performed in this study were subjected to statistical evaluation. The effects of the treatment were assessed using one-way analysis of variance (ANOVA), followed by post hoc Tukey’s tests to identify significant differences between mean values. The results are expressed as mean ± standard deviation (SD), and differences were considered statistically significant at *p*-values < 0.05. All statistical analyses were performed using STATISTICA 12 software (TIBCO Software Inc., Palo Alto, CA, USA).

## 3. Results

### 3.1. Physicochemical Characterization

#### 3.1.1. Proximate Analysis

The proximate composition of *Tetradesmus obliquus* biomass was evaluated to determine the contents of VM, A, and FC under different conditions, culture media, and collection periods. The analysis revealed distinct variations among the six samples analyzed. VM ranged from 64.80% to 72.44%, while ash varied between 6.75% and 16.77%. FC values of between 15.77% and 21.23% were recorded. These values are summarized in Table 2.

Table 2 shows the immediate proximate analysis of the crude and residual biomass of *Tetradesmus obliquus* in different cultivation media. The MV content was significantly higher (*p* < 0.05) in the crude and residual samples in the swine culture medium (2018), with 72.44 and 72.02%, respectively. This was followed by the crude biomass in the swine culture medium (2023), with 71.69%. However, after the lipid fraction was removed, the MV content of the residual biomass in the swine culture medium (2023) decreased significantly (*p* < 0.05) to 64.80 ± 0.32%. This is reflected in the increased ash content of this residual biomass, which was significantly higher (*p* < 0.05) with 16.77 ± 0.29%. The FC content was significantly higher (*p* < 0.05) in microalgae biomass samples from swine cultivation medium (2018) (21.73–20.75%), followed by microalgae biomass samples from swine cultivation medium (2023) (19.06–18.43%) and microalgae biomass samples from CHU cultivation medium (16.68–15.77%). There were no significant differences in the FC content between samples before and after the lipid fraction was removed. There was also no statistical difference in the proximate analysis between the crude and residual samples in swine culture medium (2018) after the lipid fraction was removed.

#### 3.1.2. Calorific Value

The calorific analysis of *Tetradesmus obliquus* biomass revealed variations in the HHV and LHV among the determined samples. The HHV and LHV values were significantly higher (*p* < 0.05) in the raw and residual samples in the swine culture medium (2018). There was no statistical difference in the calorific value between the raw and residual microalgae biomass samples in the swine culture medium (2018) and CHU culture medium samples after the lipid fraction was removed. However, the HHV and LHV calorific values of the crude sample in the pig culture medium (2018) were significantly different (*p* < 0.05) to those of the residual sample. This may be due to the observed changes in the volatile material and ash contents in the proximate analysis results. This may explain the decrease in the calorific value of these samples when the lipids were removed, as presented in Table 3.

#### 3.1.3. Ultimate Analysis

Table 4 presents the results of the elemental analysis performed. Given that microalgal biomass possesses unique chemical and physical characteristics that differ significantly from those of conventional fuels, this study considered the specific elemental composition of microalgae, including the carbon (C), hydrogen (H), oxygen (O), and nitrogen (N) contents. The aim is to assess the feasibility of applying microalgal biomass to various energy conversion pathways.

The carbon (C) content was significantly higher (*p* < 0.05) in the crude samples from swine cultivation medium (periods: 2018 and 2023), with 48.90% and 48.35%, respectively. After the lipid fraction was removed, the C content of the residual biomass was lower than that of the crude samples. The H content was significantly higher (*p* < 0.05) in the residual sample from swine cultivation medium (2018), with 7.85 ± 0.08%. There was no significant difference in the H content for the other samples before or after the lipid fraction was removed. The O content was significantly higher (*p* < 0.05) in the CHU residual samples (2023), with 45.19 ± 0.03%. After the lipid fraction was removed, the O content of the residual biomass was higher than that of the crude samples. The nitrogen (N) content of *Tetradesmus obliquus* biomass varied between 6.81% and 7.56%, with the N content being significantly higher (*p* < 0.05) in the residual sample from swine cultivation medium (2023). These variations reflect compositional differences influenced by the cultivation medium, lipid composition, and collection period.

### 3.2. Antioxidant Activity via DPPH

Calibration curve: Figure 8 shows the standard calibration curve for Trolox, used to determine the antioxidant activity of the microalgal biomass extracts. A strong linear correlation was observed between absorbance and Trolox concentration (R^2^ = 0.99).

Antioxidant Activity Assays: Based on the spectrophotometric assessment of antioxidant activity using the DPPH method, no significant changes were observed in the absorbance patterns of any of the biomass samples tested in the presence of DPPH, as shown in Table 5.

No antioxidant activity was detected in any of the samples when expressed in Trolox equivalents under the evaluated conditions. The spectrophotometric analysis was carried out using the DPPH assay, with absorbance readings taken at 517 nm. All samples were prepared according to the described protocol and exhibited absorbance values comparable to the negative control. This result indicates the absence of compounds capable of scavenging DPPH radicals under the experimental conditions applied, and therefore, no measurable antioxidant activity was expressed.

### 3.3. Protein Determination

The protein content of the dried *Tetradesmus obliquus* biomass was determined using the Kjeldahl and Dumas methods (via elemental analysis through combustion), with the results presented in Table 6. The total nitrogen percentage obtained by the Kjeldahl method ranged from 5.08% to 10.78%, corresponding to protein contents of between 25.78% and 54.74%. The highest nitrogen and protein values via the Kjeldahl method were observed in sample 1 (residual—swine effluent—2018), whereas the lowest values were recorded in sample 6 (crude—CHU medium—2023).

The Dumas method yielded nitrogen values ranging from 6.81% to 7.56%, resulting in estimated protein contents from 34.57% (sample 6, crude—CHU—2023) to 38.38% (sample 3, residual—swine effluent—2023), respectively. Compared to the Kjeldahl results, the nitrogen and protein contents obtained by elemental analysis exhibited less variability among the samples.

The protein values derived from elemental nitrogen were consistently lower than those estimated using the Kjeldahl method. This discrepancy may be attributed to methodological differences, particularly the broader detection of nitrogenous compounds by Kjeldahl digestion, which does not discriminate between protein and non-protein nitrogen.

These results confirm the influence of the analytical methodology on protein estimation and highlight the importance of selecting an appropriate nitrogen-to-protein conversion factor based on the specific characteristics of the analyzed biomass.

## 4. Discussion

The characterization of microalgal biomass produced in patented industrial photobioreactors contributes to ongoing efforts to explore new sources of energy generation and other industrial biotechnological applications. Microalgae have become key cultures for the global food and beverage industries, aquaculture, and both animal and human nutrition. This relevance is attributed to (i) their high content of proteins, essential amino acids, vitamins, antioxidants, omega-3 fatty acids, and minerals; (ii) their long-term sustainability, due to lower carbon, water, and arable land footprints compared to conventional crops; (iii) their role in environmental pollution remediation; and (iv) their higher productivity relative to terrestrial crops and traditional animal feed [26].

The use of microalgae as a protein source has gained increasing attention due to their high photosynthetic efficiency and nutritional profile. Studies have shown that microalgae can be incorporated into a wide range of food products, such as dietary supplements, meat substitutes, baked goods, and beverages, providing additional nutritional benefits such as dietary fiber and natural antioxidants [10]. Moreover, microalgae have emerged as a sustainable alternative for the agribusiness sector, particularly in the production of biostimulants and biofertilizers [27]. *Tetradesmus obliquus* is recognized for its potential in biofuel production due to its high lipid content and photosynthetic efficiency [28], reinforcing the growing interest in microalgae as a promising source for bioenergy and other biotechnological applications.

The thermochemical properties of microalgae are directly influenced by their VM, A, and FC content, which are essential parameters for evaluating their potential as biofuels. Volatile matter, in particular, has a significant impact on the HHV, as higher VM content tends to correlate with greater energy output [18]. The results of this study are consistent with the existing literature on dried microalgal biomass. Zakaria et al. (2022) [29] reported that the biomass of *Limnospira* (formerly *Spirulina*) sp. (Cyanobacteria) exhibited 68.15% VM, 8.06% A, and 12.57% FC.

Elemental analysis revealed significant variations in the chemical composition of the samples cultivated under different conditions. The carbon content was higher in the crude biomass samples, particularly in the 2018 batch, indicating a greater energy potential compared to the residual biomass. The increased hydrogen content observed in the samples cultivated with swine effluent in 2023 suggests a favorable contribution to the calorific value, despite a slight increase in the oxygen content, which may reduce the energy density of the biomass [30].

The HHV of biomass is positively correlated with its carbon and hydrogen contents and negatively correlated with its oxygen content. This implies that biomass with a higher oxygen content tends to exhibit a lower HHV [30,31]. Moreover, the distinction between crude and residual biomass highlights the impact of lipid extraction on energy content, with the crude biomass consistently exhibiting higher values, although the differences were not statistically significant.

These results provide a foundation for understanding the functional behavior of the biomass in industrial applications. Such characteristics, associated with the presence of organic and inorganic compounds, may directly influence oxidative stability and the preservation of bioactive fractions during biomass processing. In this context, the evaluation of the antioxidant activity of this biomass represents a complementary and strategic step, enabling the exploration of not only its energy value but also its functional potential for various applications in industrial sectors such as food, cosmetics, and pharmaceuticals [32].

The antioxidant activity of the samples was evaluated using the DPPH method, which assesses the capacity of substances to neutralize free radicals. This analysis is instrumental in determining the antioxidant potential of materials, with significant implications in various sectors, including pharmacology and food [33].

The microalga *Tetradesmus obliquus* (formerly *Scenedesmus obliquus*) (Chlorophyta) has been studied as a source of antioxidant bioactive compounds due to its rapid growth and resilience under various environmental conditions. In the DPPH method, the capacity of the extracts to reduce the DPPH radical (which has a purple color) to its neutral yellow form is measured, typically by quantifying the decrease in absorbance around 515–517 nm. Lower IC_50_ values (the extract concentration required to reduce 50% of the DPPH radical) indicate greater antioxidant potency [33,34].

According to the observations in Table 5, no antioxidant activity was expressed in terms of TE in any of the samples evaluated, with no indicative signs of such activity under the experimental conditions assessed. The samples were prepared according to the conditions described and all exhibited absorbance values similar to those of the negative control, indicating the absence of compounds with antioxidant activity.

Some probable factors may explain these negative results. *Tetradesmus obliquus* may contain antioxidants primarily in the form of trans-carotenoids and other compounds that are less reactive with DPPH [35]. Moreover, the cultivation conditions may not have favored the synthesis of polyphenols or other potent antioxidant compounds. The analyzed biomass may not contain appreciable amounts of free radical scavengers. For instance, cultures in the exponential growth phase or those grown under nutrient-rich conditions may prioritize growth-related processes (such as protein synthesis and the production of photosynthetic pigments for light capture) over the production of antioxidant secondary metabolites [36]. Therefore, it is plausible that, under the evaluated conditions, the samples lacked sufficient quantities—or the appropriate type—of antioxidants capable of reacting with DPPH at a detectable level.

Although the DPPH assay does not involve intentional fluorescence excitation, highly pigmented samples may introduce signal noise. Moreover, recent studies have indicated that chlorophyll derivatives exhibit their own absorbance peaks and even some degree of reactivity, which may obscure DPPH results when present at high concentrations. Reports have documented interference between chlorophyll derivatives and the DPPH radical at 515 nm, particularly at elevated pigment levels, which can complicate data interpretation [37]. Accordingly, it is worth noting that the samples evaluated in this study exhibited an intense green coloration, a factor that supports the findings of Alotaiby et al. (2024) [37] regarding the DPPH method and spectrophotometric wavelength readings.

The Fluorescence Resonance Energy Transfer (FRET) effect is less commonly observed, yet it cannot be ruled out. Some form of energy transfer may occur between molecules within the extract. Classical FRET requires a fluorescent donor and an acceptor whose absorption band overlaps with the donor’s emission. Chlorophyll’s fluoresce is in the red region (~680 nm); DPPH itself is not fluorescent, but its reduction products or other impurities could potentially interact. Although the literature does not explicitly report FRET between chlorophyll and DPPH, it is reasonable to extrapolate that emission or energy transfer phenomena in highly pigmented samples may indirectly disturb the radical’s equilibrium [36,37,38].

The lack of significant antioxidant activity in the evaluated samples suggests the need for further research. For instance, alternative methods for extracting antioxidant compounds, absorbance measurements at different wavelengths, or testing under varying cultivation conditions, such as analyzing the liquid inoculum before transfer into industrial-scale PBRs, could be explored. Such approaches may help to elucidate the antioxidant potential of this biomass.

The composition of algal biomass is often the primary driver of the economic viability of research and the development of commercial algae-based products. As the field of commercial algae production continues to expand, there is a growing need for standardized terminology between producers and markets with regard to biomass characterization [39].

Cultivating microalgae in nitrogen-deficient media is considered one of the most effective strategies for increasing their lipid content. Nitrogen limitation enhances lipid accumulation while altering overall biomass productivity. In fact, microalgae are known to modify their biochemical composition in response to changing environmental conditions. The nitrogen concentration in the culture medium directly influences biomass composition, affecting both the growth rate and the proportion of stored macromolecules [40,41].

The study by Akgül (2024) [40] demonstrated that under nitrogen-deficient conditions, a decrease in the specific growth rate and protein content of *Tetradesmus obliquus* was observed, while the lipid content increased. Protein levels ranged between 47.62% and 47.82% of dry weight in media containing 75% and 50% of the available nitrogen, respectively. In addition, Vladić et al. (2023) [42], in research on the application of *T. obliquus* for wastewater remediation, demonstrated its ability to absorb nitrogenous compounds and synthesize proteins from these nutrients.

The determination of the protein content in biomass samples, such as microalgae, is commonly carried out using the Kjeldahl and Dumas methods. Both techniques measure the total nitrogen content, but they differ in the analytical procedures employed [43].

The Kjeldahl method is widely recognized and involves three main steps: digestion, distillation, and back titration. It is considered the official method for protein determination in food by the Association of Official Analytical Chemists (AOAC) International. This method relies on the conversion of organic nitrogen present in the sample into ammonia, which is then quantified. In contrast, the Dumas method involves the combustion of samples, releasing nitrogen in its gaseous form, which is directly quantified by a thermal conductivity detector after the removal of interfering substances such as CO_2_ and water [43].

A significant advantage of the Dumas method over the Kjeldahl method is its speed and reduced use of hazardous reagents, as well as the requirement for smaller sample quantities. However, both methods may be subject to interference, potentially leading to divergent results, particularly in the presence of non-protein nitrogen compounds, which affect the nitrogen-to-protein conversion factor (k), typically 6.25, and may result in the under- or overestimation of the actual protein content [43]. It is worth noting that, for the present study, a conversion factor (k) of 5.08 was applied, as recommended by Templeton and Laurens (2015) [22] for the *Tetradesmus* genus.

In the study conducted by Cruz et al. (2018) [44], the protein content of *Monoraphidium* sp. (Chlorophyta) biomass, cultivated in a 100 L photobioreactor under controlled conditions, was determined using the Kjeldahl method following centrifugation and lyophilization. The authors reported a protein content of 34.26% in the evaluated biomass.

In a review on microalgal proteins, Xu et al. (2024) [45] reported that some species may exhibit protein contents exceeding 60% on a dry weight basis, particularly when cultivated under optimal nitrogen availability. Moreover, other studies have indicated that the protein content of *Tetradesmus obliquus* biomass can vary widely depending on nutrient limitation and the extraction strategies employed [46,47].

The study conducted by Lane et al. (2021) [39] investigated the elemental composition and protein content of two microalgae species, *Nannochloropsis salina* and *Scenedesmus acutus LRB-AP-0401*. The characterization involved determining ash, elemental carbon, and nitrogen, as well as estimating the protein content based on the measured nitrogen levels. The reported protein contents were 40.92% for *S. obliquus* and 31.98% for *N. salina*. These differences were primarily attributed to the application of different nitrogen-to-protein conversion factors specific to each species, 6.25 for *S. obliquus* and 4.78 for *N. salina*, taking into account the particularities of the amino acid composition of each microalga.

In summary, the protein averages determined in the present study ranged from 40.02% (Kjeldahl) to 36.66% (Dumas). These values are consistent with the ranges commonly reported in the literature for microalgae. It is important to emphasize that the biomass analyzed in this study was produced on an industrial scale and subjected to various physicochemical stress conditions throughout the production process, including exposure to exhaust gases, climatic fluctuations, flocculation, and high temperatures. Nevertheless, the protein contents obtained remained in line with the literature values, reinforcing the biotechnological and industrial applicability of the biomass. Furthermore, based on the protein determinations presented in this study using both the Kjeldahl and Dumas methods, it is possible to apply the more commonly adopted nitrogen-to-protein conversion factor (6.25) to obtain alternative results. Doing so would yield values even higher than those reported here.

Overall, the variations observed in the biochemical composition of the evaluated biomass can be attributed to the combined effects of nutrient availability and cultivation duration, both of which play a crucial role in regulating cellular metabolism. Under nutrient-limited conditions, microalgae typically shift their metabolic pathways towards the accumulation of storage compounds such as lipids and carbohydrates, often reducing protein synthesis as a physiological response. In contrast, nitrogen-rich environments tend to promote protein accumulation and faster biomass growth. Additionally, the length of the cultivation period influences the specific growth phase reached by the microalgae, with each phase associated with distinct metabolic activity and resource allocation strategies, ultimately shaping the final biochemical profile of the biomass [41,47].

It is important to note that the complexity of the cultivation media used in this study, especially those derived from swine manure, introduces additional variability in biomass composition. The swine manure collected over different production cycles inevitably reflects differences in the animal feed composition, growth stages, and physiological conditions of the livestock at the time of waste generation. These factors contribute to variations in the nutrient content and organic load within the culture media, leading to additional metabolic adjustments by the microalgae during growth.

## 5. Conclusions

This study provides comprehensive insights into the biochemical and thermochemical properties of industrially cultivated *Tetradesmus obliquus*, highlighting its potential as a sustainable resource for bioenergy and other industrial applications. Proximate analysis was essential for establishing key parameters to assess the biomass as a solid biofuel, while the calorific value results confirmed its viability as an energy source due to its favorable energy density. Ultimate analysis provided critical elemental composition data, particularly the nitrogen content, which enabled accurate protein estimation. Both the Kjeldahl and Dumas methods yielded protein values consistent with those reported in the literature, reinforcing the suitability of *T. obliquus* biomass for use in industries focused on protein-rich raw materials and sustainable bioproducts.

Although no significant antioxidant activity was detected under the experimental DPPH conditions, this finding suggests the need for further studies exploring alternative extraction methods, cultivation conditions, or antioxidant assays to fully assess the functional potential of this biomass. Overall, the multiparametric characterization presented here strengthens the scientific basis for scaling up *T. obliquus* biomass production and opens avenues for its applications in the food, pharmaceutical, and bioenergy sectors, contributing to the development of circular and low-carbon industrial processes.

## Figures and Tables

**Figure 1 microorganisms-13-01583-f001:**
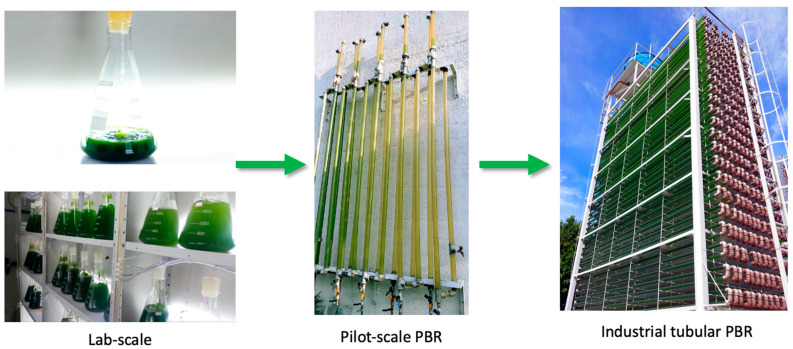
Photobioreactors at NPDEAS, Federal University of Paraná (Curitiba, Brazil).

**Figure 2 microorganisms-13-01583-f002:**
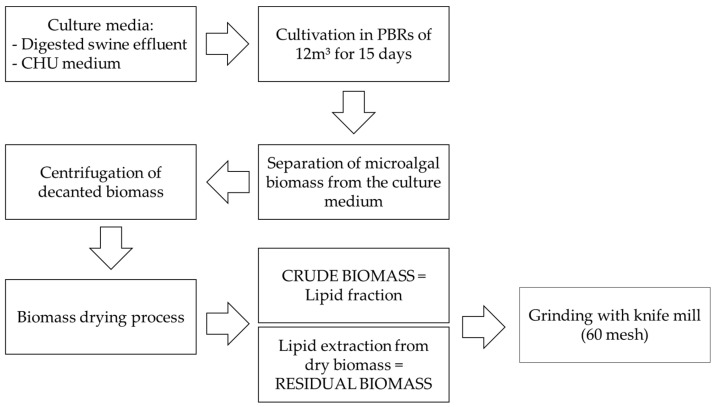
Schematic representation of microalgal biomass extraction.

**Figure 3 microorganisms-13-01583-f003:**
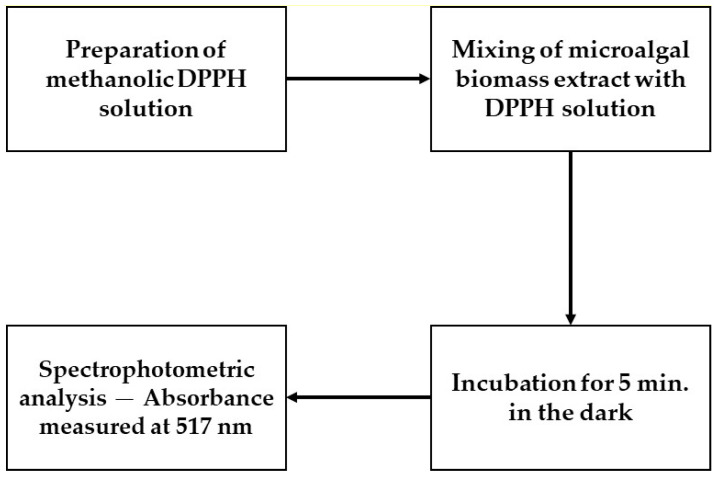
Experimental workflow for antioxidant activity assessment.

**Figure 4 microorganisms-13-01583-f004:**
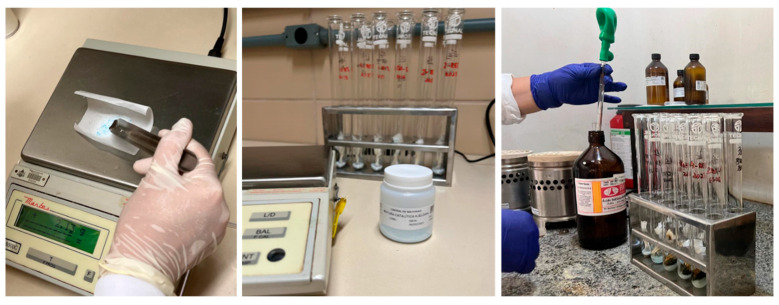
Preparation of digestion tubes.

**Figure 5 microorganisms-13-01583-f005:**
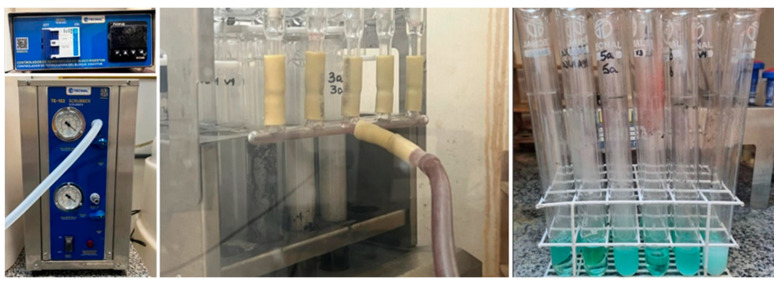
Digestion process.

**Figure 6 microorganisms-13-01583-f006:**
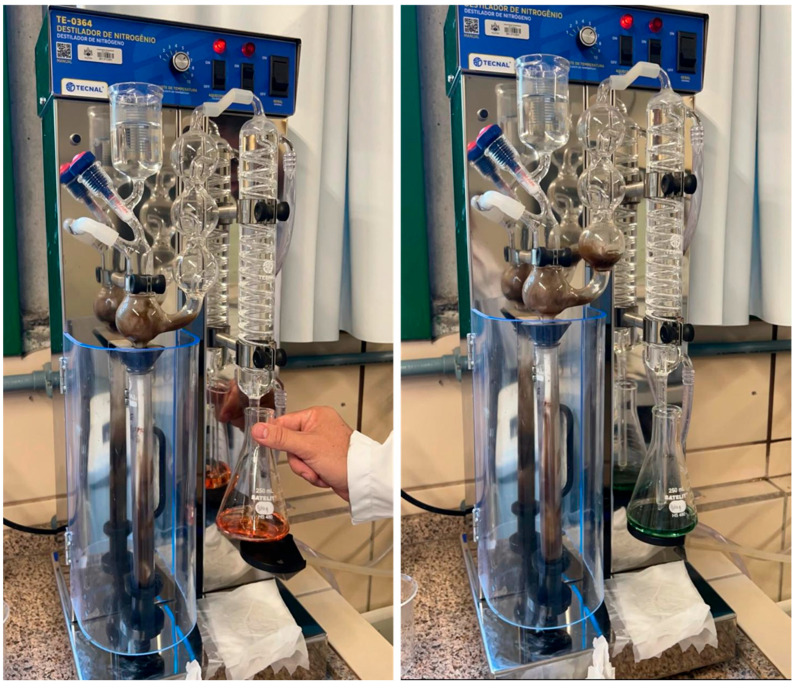
Distillation process.

**Figure 7 microorganisms-13-01583-f007:**
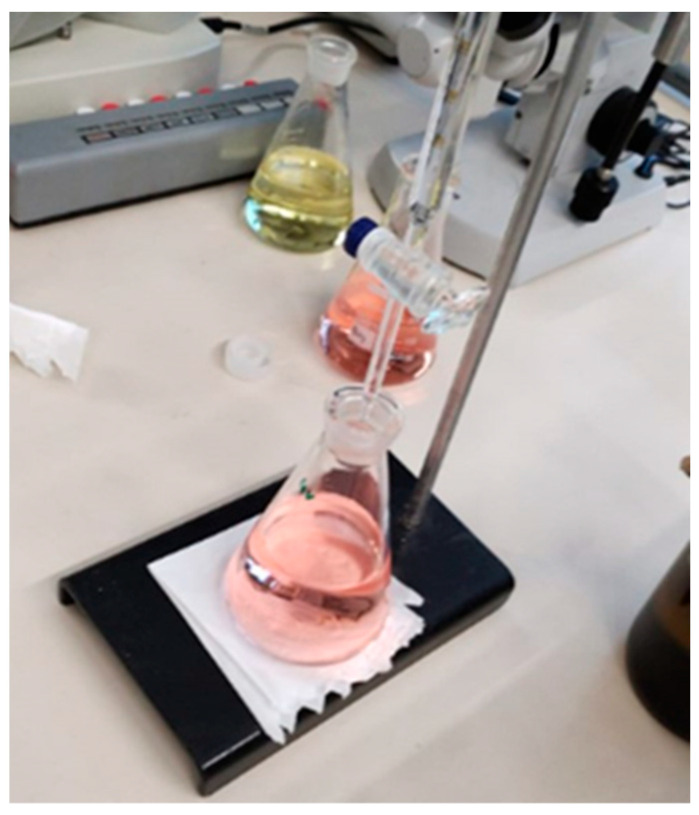
Titration process.

**Figure 8 microorganisms-13-01583-f008:**
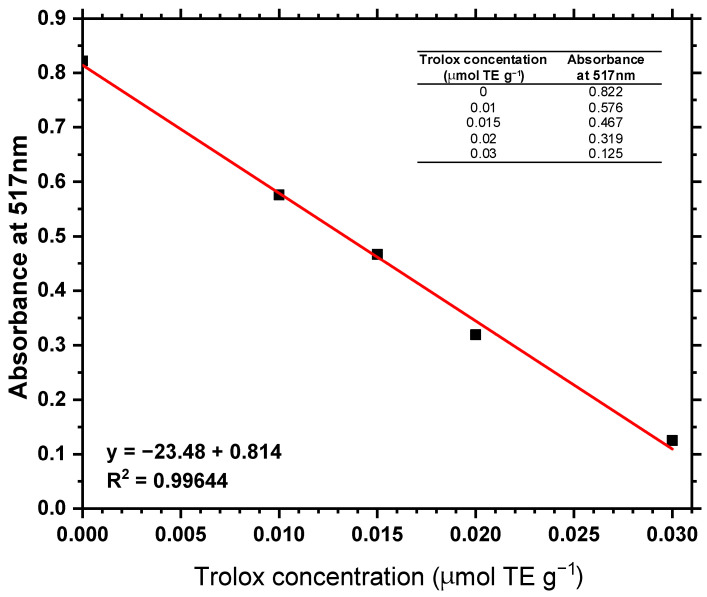
Calibration curve in Trolox equivalents (µmol TE·g^−1^).

**Table 1 microorganisms-13-01583-t001:** Overview of experimental sample conditions.

Sample	Biomass Description	Growing Medium	Acquisition Period
1	Residual dried	Digested swine effluent	2018
2	Crude dried **^1^**	Digested swine effluent	2018
3	Residual dried	Digested swine effluent	2023
4	Crude dried **^1^**	Digested swine effluent	2023
5	Residual dried	CHU medium	2023
6	Crude dried **^1^**	CHU medium	2023

**^1^** Crude biomass refers to the lipid-rich fraction of microalgal biomass before lipid extraction was performed.

**Table 2 microorganisms-13-01583-t002:** Proximate analysis of *Tetradesmus obliquus* biomass under different cultivation conditions.

Sample	VM (%)		A (%)		FC (%)	
	$\bar{x}$	s	$\bar{x}$	s	$\bar{x}$	s
1: Residual—swine effluent—2018	72.02	±0.23 **a**	6.75	±0.23 **e**	21.23	±0.05 **a**
2: Crude—swine effluent—2018	72.44	±0.13 **a**	6.81	±0.05 **e**	20.75	±0.14 **a**
3: Residual—swine effluent—2023	64.80	±0.32 **d**	16.77	±0.29 **a**	18.43	±0.17 **b**
4: Crude—swine effluent—2023	71.69	±0.33 **a**	9.25	±0.36 **d**	19.06	±0.12 **b**
5: Residual—CHU medium—2023	69.15	±1.01 **b**	15.08	±0.23 **c**	15.77	±0.81 **c**
6: Crude—CHU medium—2023	67.52	±0.55 **c**	15.80	±0.28 **b**	16.68	±0.27 **c**

VM = volatile matter; A = ash; FC = fixed carbon. $\bar{x}\;$ = mean; s = standard deviation. Bold lower-case letters followed by the same letter within a column indicate no significant difference between components at a 5% significance level according to Tukey’s test. Results are expressed on a dry weight basis.

**Table 3 microorganisms-13-01583-t003:** Calorific value of *Tetradesmus obliquus* biomass (dry basis).

Sample	HHV _db_ (MJ·kg^−1^)		LHV _db_ (MJ·kg^−1^)	
	$\bar{x}$	s	$\bar{x}$	s
1: Residual—swine effluent—2018	22.56	±0.02 **a**	21.09	±0.02 **a**
2: Crude—swine effluent—2018	22.75	±0.05 **a**	21.24	±0.05 **a**
3: Residual—swine effluent—2023	17.53	±0.15 **d**	15.91	±0.15 **d**
4: Crude—swine effluent—2023	20.38	±0.08 **b**	18.87	±0.08 **b**
5: Residual—CHU medium—2023	18.32	±0.05 **c**	16.86	±0.05 **c**
6: Crude—CHU medium—2023	18.37	±0.09 **c**	16.87	±0.09 **c**

HHV = higher heating value; LHV = lower heating value. $\bar{x}\;$ = mean; s = standard deviation. db = dry basis. Bold lower-case letters followed by the same letter within a column indicate no significant difference between components at a 5% significance level according to Tukey’s test.

**Table 4 microorganisms-13-01583-t004:** Elemental composition of *Tetradesmus obliquus* biomass.

Sample	C (%)		H (%)		N (%)		O (%)	
	$\bar{x}$	s	$\bar{x}$	s	$\bar{x}$	s	$\bar{x}$	s
1: Residual—swine effluent—2018	47.11	±0.02 **b**	7.13	±0.01 **b**	7.12	±0.02 **c**	38.64	±0.05 **c**
2: Crude—swine effluent—2018	48.90	±0.24 **a**	7.32	±0.14 **b**	7.18	±0.02 **bc**	36.60	±0.39 **d**
3: Residual—swine effluent—2023	45.98	±0.20 **c**	7.85	±0.08 **a**	7.56	±0.12 **a**	38.61	±0.40 **c**
4: Crude—swine effluent—2023	48.35	±0.12 **a**	7.33	±0.12 **b**	7.21	±0.05 **bc**	37.11	±0.30 **d**
5: Residual—CHU medium—2023	40.74	±0.00 **e**	7.26	±0.01 **b**	6.81	±0.02 **d**	45.19	±0.03 **a**
6: Crude—CHU medium—2023	42.88	±0.22 **d**	7.09	±0.11 **b**	7.43	±0.07 **ab**	42.60	±0.40 **b**

C = carbon; H = hydrogen; N = nitrogen; O = oxygen. $\bar{x}\;$ = mean; s = standard deviation. Bold lower-case letters followed by the same letter within a column indicate no significant difference between components at a 5% significance level according to Tukey’s test. Results are expressed on an ash-free dry basis.

**Table 5 microorganisms-13-01583-t005:** Spectrophotometric absorbance of microalgal biomass in DPPH radical scavenging assay.

Sample—Biomass Source Conditions	Sample Solution Volume (mL)	DPPH (mM)	Absorbance (nm)
1: Residual—swine effluent—2018 **^MSF^**	0.2	60	0.771
2: Crude—swine effluent—2018 **^MSF^**	0.2	60	0.948
3: Residual—swine effluent—2023 **^MSF^**	0.2	60	0.956
4: Crude—swine effluent—2023 **^MSF^**	0.2	60	0.953
5: Residual—CHU medium—2023 **^MSF^**	0.2	60	0.966
6: Crude—CHU medium—2023 **^MSF^**	0.2	60	0.912

MSF = methanol-soluble fraction.

**Table 6 microorganisms-13-01583-t006:** Protein estimation of microalgal biomass based on nitrogen content using different analytical methods.

Sample	%N Kjeldahl		% Protein		%N Elemental		% Protein	
	$\bar{x}$	s	$\bar{x}$	s	$\bar{x}$	s	$\bar{x}$	s
**1**	10.78	±2.18 **a**	54.74	±11.06 **a**	7.12	±0.02 **c**	36.17	±0.07 **c**
**2**	8.70	±0.87 **ab**	44.22	±4.43 **ab**	7.18	±0.02 **bc**	36.50	±0.11 **bc**
**3**	8.50	±0.21 **ab**	43.18	±1.05 **ab**	7.56	±0.12 **a**	38.38	±0.61 **a**
**4**	7.23	±0.39 **ab**	36.74	±1.97 **ab**	7.21	±0.05 **bc**	36.63	±0.29 **bc**
**5**	6.98	±0.39 **ab**	35.48	±1.99 **ab**	6.81	±0.02 **d**	37.74	±0.36 **d**
**6**	5.08	±0.58 **b**	25.78	±2.96 **b**	7.43	±0.07 **ab**	34.57	±0.11 **ab**

**1**: Residual biomass—swine effluent—2018; **2**: crude biomass—swine effluent—2018; **3**: residual biomass—swine effluent—2023; **4**: crude biomass—swine effluent—2023; **5**: residual biomass—CHU medium—2023; **6**: crude biomass—CHU medium—2023. $\bar{x}\;$ = mean; s = standard deviation. Bold lower-case letters followed by the same letter within a column indicate no significant difference between components at a 5% significance level according to Tukey’s test.

## Data Availability

The original contributions presented in this study are included in the article. Further inquiries can be directed to the corresponding authors.

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
