# Peer review of "Multiparametric Evaluation of Tetradesmus obliquus Biomass: An Integrated Approach Including Antioxidant, Nutritional, and Energy Properties"

_microorganisms, 2025, doi:10.3390/microorganisms13071583_

Round 1

Reviewer 1 Report

Comments and Suggestions for Authors

Comments on manuscript entitled "Multiparametric evaluation of Tetradesmus obliquus biomass: An integrated approach of antioxidant, nutritional, and energy properties"

The manuscript assessed the nutritional, antioxidant, and energy potential of Tetradesmus obliquus. This article provides an interesting and important topic that has received an attention in the recent few years. However, it requires minor revision before it can be considered for publishing.

-  Add more quantitative results in the abstract section.

- Clearly define the aim of the manuscript in the introduction section.

- Statistical analysis method should be given in materials and methods section.

- In table 2, you can replace "MV" with "VM".

- Table 2, 3, 4: MV, A, FC, HHV, LHV … etc. are abbreviations and their full names should be written in the footnote.

- The term "statistically significant or non-significant" must be used clear in the Results and Discussion section.

- Despite comparisons of different culture media and periods, the mechanistic interpretation of how these conditions affect biomass composition remains underdeveloped. Please explain.

- The conclusions section should be improved by highlighting the findings of this manuscript and the potential application of this research in the future.

Author Response

Manuscript ID - microorganisms-3683731

Point-by-point response to the reviewer’s comments

Reviewer 1:

The manuscript assessed the nutritional, antioxidant, and energy potential of Tetradesmus obliquus. This article provides an interesting and important topic that has received an attention in the recent few years. However, it requires minor revision before it can be considered for publishing.

R: We thank the reviewer for the positive and encouraging feedback on the relevance of our manuscript. We appreciate the recognition of the importance of the topic and have carefully addressed all suggestions for improvement.

-  Add more quantitative results in the abstract section.

R: We appreciate the reviewer’s suggestion to include more quantitative data in the abstract. Accordingly, we have added key numerical results, including the ranges of volatile matter (64.80–72.44%) and fixed carbon (15.77–21.23%), as well as the higher and lower heating values (HHV: 18.32–22.75 MJ·kg⁻¹; LHV: 16.86–21.24 MJ·kg⁻¹), to provide a more concise summary of the biomass’s energy potential.

- Clearly define the aim of the manuscript in the introduction section.

R: We thank the reviewer for this valuable suggestion. To address the comment, we have revised the final paragraphs of the Introduction to clearly state the aim and scope of the study. The revised section now reads: “Given this context, the present study aimed to perform a comprehensive multiparametric characterization of T. obliquus biomass, focusing on its physicochemical, nutritional, thermochemical, and antioxidant properties. This characterization was conducted to evaluate the potential as an innovative ingredient for industrial applications, particularly within the food, pharmaceutical, and bioenergy sectors. To this end, dry biomass samples produced at the Sustainable Energy Research and Development Center (NPDEAS) research group were analyzed.

The biomass was cultivated in a patented industrial-scale PBR system integrated with a residues incinerator, enabling large-scale production under controlled engineering conditions. This sustainable production process aligns with several United Nations Sustainable Development Goals (SDGs), including: Goal 2 – Zero Hunger; Goal 3 – Good Health and Well-being; Goal 7 – Affordable and Clean Energy; Goal 9 – Industry, Innovation and Infrastructure; Goal 11 – Sustainable Cities and Communities; Goal 12 – Responsible Consumption and Production; and Goal 13 – Climate Action.”

We believe this clarification strengthens the contextual framework and reinforces the relevance of the study objectives.

- Statistical analysis method should be given in materials and methods section.

R: We thank the reviewer for this important observation. To address this point, we have added a new paragraph at the end of the Materials and Methods section, as follows: “All experimental data obtained from the different analyses performed in this study were subjected to statistical evaluation. The effects of treatment were assessed using one-way analysis of variance (ANOVA), followed by post hoc Tukey tests to identify significant differences between mean values. Results are expressed as mean ± standard deviation (SD), and differences were considered statistically significant at p-values < 0.05. All statistical analyses were performed using STATISTICA 12 software (TIBCO Software Inc., USA).”

- In table 2, you can replace "MV" with "VM".

R: We have replaced it in Table 2. Thanks.

- Table 2, 3, 4: MV, A, FC, HHV, LHV … etc. are abbreviations and their full names should be written in the footnote.

R: These changes have been made. Thanks for the suggestion.

- The term "statistically significant or non-significant" must be used clear in the Results and Discussion section.

R: We appreciate the reviewer’s recommendation. In response, we have revised the relevant paragraphs throughout the Results and Discussion section to clearly state whether observed differences were statistically significant or non-significant, based on the ANOVA and Tukey test results (p < 0.05). The terminology has been standardized to improve clarity and scientific accuracy. Please see examples of the revised text below:

“Table 2 shows the immediate proximate analysis of the crude and residual biomass of Tetradesmus obliquus in different cultivation media. The MV content was significantly higher (p < 0.05) in the crude and residual samples in the swine culture medium (2018), with 72.44 and 72.02%, respectively. This was followed by the crude biomass in the swine culture medium (2023), with 71.69%. However, after the lipid fraction was removed, the MV content of the residual biomass in the swine culture medium (2023) decreased significantly (p < 0.05) to 64.80 ± 0.32%. This is reflected in the increased ash content of this residual biomass, which was significantly higher (p < 0.05) with 16.77 ± 0.29%. The FC content was significantly higher (p < 0.05) in microalgae biomass samples from swine cultivation (2018) (21.73–20.75%), followed by microalgae biomass samples from swine cultivation (2023) (19.06–18.43%) and microalgae biomass samples from CHU cultivation medium (16.68–15.77%). There were no significant differences in the FC content between samples before and after the lipid fraction was removed. There was also no statistical difference in the proximate analysis between the crude and residual samples in swine culture (2018) after the lipid fraction was removed.”

“The calorific analysis of Tetradesmus obliquus biomass revealed variations in HHV and LHV among the determined samples. The HHV and LHV values were significantly higher (p < 0.05) in the raw and residual samples in the swine culture medium (2018). There was no statistical difference in calorific value between the raw and residual in microalgae biomass samples, the swine culture medium (2018), and CHU culture medium samples after the lipid fraction was removed. However, the HHV and LHV calorific values were significantly different (p < 0.05) in the crude sample in the pig culture medium (2018) than in the residual sample. This may be due to the observed changes in volatile material and ash contents in the proximate analysis results. This may explain the decrease in the calorific value of these samples when the lipids are removed. As presented in Table 3.”

“The carbon (C) content was significantly higher (p < 0.05) in the crude samples from swine cultivation (periods: 2018 and 2023), with 48.90% and 48.35%, respectively. After the lipid fraction was removed, the C content of the residual biomass was lower than that of the crude samples. The H content was significantly higher (p < 0.05) in the residual sample from swine cultivation (2018), with 7.85 ± 0.08%. There was no significant difference in H content for the other samples before or after the lipid fraction was removed. The O content was significantly higher (p < 0.05) in the CHU residual samples (2023), with 45.19 ± 0.03%. After the lipid fraction was removed, the O content of the residual biomass was higher than that of the crude samples. The nitrogen (N) content of Tetradesmus obliquus biomass varied between 6.81% and 7.56%, with the N content being significantly higher (p < 0.05) in the residual sample from swine cultivation (2023). These variations reflect compositional differences influenced by cultivation medium, lipid composition, and collection period.”

- Despite comparisons of different culture media and periods, the mechanistic interpretation of how these conditions affect biomass composition remains underdeveloped. Please explain.

R: We appreciate the reviewer’s insightful observation. To address this point, we have expanded the Results and Discussion section by including a mechanistic interpretation of how cultivation conditions influence the biochemical composition of T. obliquus biomass. The following paragraphs were added:

“Overall, the variations observed in the biochemical composition of the evaluated biomass can be attributed to the combined effects of nutrient availability and cultivation duration, both of which play a crucial role in regulating cellular metabolism. Under nutrient-limited conditions, microalgae typically shift their metabolic pathways towards the accumulation of storage compounds such as lipids and carbohydrates, often reducing protein synthesis as a physiological response. In contrast, nitrogen-rich environments tend to promote protein accumulation and faster biomass growth. Additionally, the length of the cultivation period influences the specific growth phase reached by the microalgae, with each phase associated with distinct metabolic activity and resource allocation strategies, ultimately shaping the final biochemical profile of the biomass [48, 49].

It is important to note that the complexity of the cultivation media used in this study, especially those derived from swine manure, introduces additional variability in biomass composition. The swine manure collected over different production cycles inevitably reflects differences in animal feed composition, growth stages, and physiological conditions of the livestock at the time of waste generation. These factors contribute to variations in nutrient content and organic load within the culture media, leading to additional metabolic adjustments by the microalgae during growth.”

- The conclusions section should be improved by highlighting the findings of this manuscript and the potential application of this research in the future.

R: In response to the reviewer’s suggestion, the following paragraph has been changed to:

“This study provides comprehensive insights into the biochemical and thermochemical properties of industrially cultivated Tetradesmus obliquus, highlighting its potential as a sustainable resource for bioenergy and other industrial applications. Proximate analysis was essential for establishing key parameters to assess the biomass as a solid biofuel, while the calorific value results confirmed its viability as an energy source due to its favorable energy density. Ultimate analysis provided critical elemental composition data, particularly nitrogen content, which enabled accurate protein estimation. Both Kjeldahl and Dumas methods yielded protein values consistent with those reported in the literature, reinforcing the suitability of T. obliquus biomass for use in industries focused on protein-rich raw materials and sustainable bioproducts.

 Although no significant antioxidant activity was detected under the experimental DPPH conditions, this finding suggests the need for further studies exploring alternative extraction methods, cultivation conditions, or antioxidant assays to fully assess the functional potential of this biomass. Overall, the multiparametric characterization presented here strengthens the scientific basis for scaling up T. obliquus biomass production and opens avenues for its applications in the food, pharmaceutical, and bioenergy sectors, contributing to the development of circular and low-carbon industrial processes.”

Reviewer 2 Report

Comments and Suggestions for Authors

The manuscript entitled "Multiparametric evaluation of Tetradesmus obliquus biomass: An integrated approach to antioxidant, nutritional and energetic properties" addresses a relevant and appropriate topic for this journal.
The manuscript is well written, well structured and well founded.
I suggest that necessary corrections be made, especially regarding the taxonomy/nomenclature of the algae cited.
Authors should always use valid names and not their synonyms.

Corrections needed:

line 140/141 -  to flocculation tanks for biomass recovery. To facilitate separation, the flocculant "Tanfloc SG" ... (Note: words written in italics are reserved for the names of genus and/or species)

line 144 -  tion. The recovered biomass was then concentrated via centrifugation using a "US Centrifuge System M512", ... (Note: words written in italics are reserved for the names of genus and/or species)

line 213/214 - the combustion process [17,18]. The analyses were performed using a "PerkinElmer 2400 Series II CHN-O" elemental analyser. (Note: words written in italics are reserved for the names of genus and/or species)

line 225 - homogenised by agitation and incubated in the dark for 5 min. Absorbance readings

line 254 - for species belonging to the genera Scenedesmus/Tetradesmus. The main steps of the adapted methodol-

line 256 -  • Moisture content determination: "Miller Toledo HE53". Since the presence of moisture (Note: words written in italics are reserved for the names of genus and/or species)

line 260 - mined using a Mettler "Toledo HE53" moisture analyser. (Note: words written in italics are reserved for the names of genus and/or species)

line 271 - The digestion was carried out using a "TE-152 – SCRUBBER" digester. The tube was  (Note: words written in italics are reserved for the names of genus and/or species)

line 273 - then increased gradually by 50 °C every 10 min until reaching 350 °C, at which point

line 287 - min or until the solution in the Erlenmeyer flask displayed a stable green colour,

line 289 -  using a "TE-0364" nitrogen distiller, as shown in Figure 6. (Note: words written in italics are reserved for the names of genus and/or species)

line 308 - "d𝑟𝑦 𝑠𝑎𝑚𝑝𝑙𝑒 𝑚𝑎𝑠𝑠" in g. (Note: words written in italics are reserved for the names of genus and/or species)

line 436 - that the biomass of Limnospira (formerly Spirulina) sp. (Cyanobacteria) exhibited 68.15% VM, 8.06% A, and 12.57% FC. 

line 462 - The microalga Tetradesmus obliquus (formerly Scenedesmus obliquus) (Chlorophyta) has been studied as a 

line 546/547 - In the study conducted by Cruz et al. (2018) [44], the protein content of Monoraphidium sp. (Chlorophyta) biomass,

line 556 - tion and protein content of the microalgae Tetradesmus obliquus (formerly Scenedesmus acutus) (Chlorophyta) and  Microchloropsis salina (formerly Nannochloropsis salina) (Eustigmatophyceae). 

line 559 - contents were 40.92% for T. obliquus and 31.98% for M. salina. These

line 561/562 - conversion factors specific to each species: 6.25 for T. obliquus and 4.78 for M. salina ...

Author Response

Manuscript ID - microorganisms-3683731

Point-by-point response to the reviewer’s comments

Reviewer 2:

The manuscript entitled "Multiparametric evaluation of Tetradesmus obliquus biomass: An integrated approach to antioxidant, nutritional and energetic properties" addresses a relevant and appropriate topic for this journal.
The manuscript is well written, well structured and well founded.
I suggest that necessary corrections be made, especially regarding the taxonomy/nomenclature of the algae cited.
Authors should always use valid names and not their synonyms.

R: We thank the reviewer for the positive and encouraging comments on the manuscript’s structure, clarity, and relevance. Regarding the taxonomic/nomenclatural recommendation, we fully agree with the importance of using valid and up-to-date scientific names. Accordingly, we have reviewed the manuscript carefully to ensure that.

Corrections needed:

line 140/141 -  to flocculation tanks for biomass recovery. To facilitate separation, the flocculant "Tanfloc SG" ... (Note: words written in italics are reserved for the names of genus and/or species)

R: We thank the reviewer for this observation. The use of italics has been removed from the text.

line 144 -  tion. The recovered biomass was then concentrated via centrifugation using a "US Centrifuge System M512", ... (Note: words written in italics are reserved for the names of genus and/or species)

R: We thank the reviewer for this observation. The use of italics has been removed from the text.

line 213/214 - the combustion process [17,18]. The analyses were performed using a "PerkinElmer 2400 Series II CHN-O" elemental analyzer. (Note: words written in italics are reserved for the names of genus and/or species)

R: We thank the reviewer for this observation. The use of italics has been removed from the text.

line 225 - homogenised by agitation and incubated in the dark for 5 min. Absorbance readings

R: We thank the reviewer for pointing out this section. To improve clarity and ensure proper scientific language, we have revised the paragraph describing the antioxidant activity assay. The updated text now reads:

“Antioxidant activity assays were conducted by adding 100 mg of each of the six microalgal biomass samples (solid fraction) to 100 mL of methanol. The mixtures were filtered to obtain the methanol-soluble fractions, from which 0.2 mL aliquots were combined with 2.8 mL of a methanolic DPPH solution (final DPPH concentration: 0.02 mM). The solutions were homogenized by agitation and incubated in the dark for 5 minutes. Absorbance was then measured at 517 nm using a spectrophotometer. Reductions in absorbance, indicative of antioxidant activity, were expressed in Trolox equivalents (TE), based on the calibration curve. A schematic overview of the experimental steps is shown in Figure 3.”

line 254 - for species belonging to the genera Scenedesmus/Tetradesmus. The main steps of the adapted methodol-

R: We thank the reviewer for noting this point. It has been changed to: “genera Scenedesmus/Tetradesmus.”

line 256 -  • Moisture content determination: "Miller Toledo HE53". Since the presence of moisture (Note: words written in italics are reserved for the names of genus and/or species)

R: We thank the reviewer for this observation. Tthe use of italics has been removed from the text where it was not appropriate.

line 260 - mined using a Mettler "Toledo HE53" moisture analyser. (Note: words written in italics are reserved for the names of genus and/or species)

R: We thank the reviewer for this observation. As requested, the use of italics has been removed from the text where it was not appropriate.

line 271 - The digestion was carried out using a "TE-152 – SCRUBBER" digester. The tube was  (Note: words written in italics are reserved for the names of genus and/or species)

R: We thank the reviewer for this observation. As requested, the use of italics has been removed from the text where it was not appropriate.

line 273 - then increased gradually by 50 °C every 10 min until reaching 350 °C, at which point

R: We thank the reviewer for pointing out this phrasing. To improve clarity and grammatical consistency, we have revised the sentence.

line 287 - min or until the solution in the Erlenmeyer flask displayed a stable green colour,

R: We have revised the sentence. Thanks.

line 289 -  using a "TE-0364" nitrogen distiller, as shown in Figure 6. (Note: words written in italics are reserved for the names of genus and/or species)

R: We appreciate the reviewer’s observation regarding this sentence. In response, it was changed.

line 308 - "d?? ?????? ????" in g. (Note: words written in italics are reserved for the names of genus and/or species)

R: We appreciate the reviewer’s observation regarding this sentence. In response, it was changed.

line 436 - that the biomass of Limnospira (formerly Spirulina) sp. (Cyanobacteria) exhibited 68.15% VM, 8.06% A, and 12.57% FC. 

R: We appreciate the reviewer’s observation regarding this sentence. In response, it was changed.

line 462 - The microalga Tetradesmus obliquus (formerly Scenedesmus obliquus) (Chlorophyta) has been studied as a 

R: We appreciate the reviewer’s observation regarding this sentence. In response, it was changed.

line 546/547 - In the study conducted by Cruz et al. (2018) [44], the protein content of Monoraphidium sp. (Chlorophyta) biomass,

R: We appreciate the reviewer’s observation regarding this sentence. In response it was changed.

line 556 - tion and protein content of the microalgae Tetradesmus obliquus (formerly Scenedesmus acutus) (Chlorophyta) and Microchloropsis salina (formerly Nannochloropsis salina) (Eustigmatophyceae).

R: We appreciate the reviewer’s observation regarding this sentence. It was changed.

line 559 - contents were 40.92% for T. obliquus and 31.98% for M. salina. These

R: We appreciate the reviewer’s observation regarding this sentence. In response, it was changed.

line 561/562 - conversion factors specific to each species: 6.25 for T. obliquus and 4.78 for M. salina ...

R: We appreciate the reviewer’s observation regarding this sentence. In response, it was changed.
